In silico analyses for potential key genes associated with gastric cancer

Yan Ping 1
He Yingchun 2
Xie Kexin 2
Kong Shan 2
Zhao Weidong wdzhao@dali.edu.cn 2
1 Department of Gastroenterology, Clinical College, Dali University , Dali , Yunnan , China
2 Department of Clinical Laboratory, Clinical College, Dali University , Dali , Yunnan , China
Uversky Vladimir
Electronic publication date: 2018 Dec 7
Publication date: 2018
Volume: 6
Electronic Location ID: e6092
Received 2018 Oct 2; Accepted 2018 Nov 9
Copyright: ©2018 Yan et al.
Copyright year: 2018
Copyright holder: Yan et al.
License: This is an open access article distributed under the terms of the Creative Commons Attribution License, which permits unrestricted use, distribution, reproduction and adaptation in any medium and for any purpose provided that it is properly attributed. For attribution, the original author(s), title, publication source (PeerJ) and either DOI or URL of the article must be cited.
License URL: https://creativecommons.org/licenses/by/4.0/

Keywords: Gastric cancer, Hub genes, In silico, Bioinformatics analysis

Funding: Doctoral Scientific Research Foundation of Dali University KYBS201721 Foundation of Yunnan Educational Committee 2017YJS021 This work was supported by the Doctoral Scientific Research Foundation of Dali University (No. KYBS201721) to WD Zhao, and the Foundation of Yunnan Educational Committee (No. 2017YJS021) to P Yan. The funders had no role in study design, data collection and analysis, decision to publish, or preparation of the manuscript.

==============================
Background

Understanding hub genes involved in gastric cancer (GC) metastasis could lead to effective approaches to diagnose and treat cancer. In this study, we aim to identify the hub genes and investigate the underlying molecular mechanisms of GC.

Methods

To explore potential therapeutic targets for GC,three expression profiles (GSE54129, GSE33651, GSE81948) of the genes were extracted from the Gene Expression Omnibus (GEO) database. The GEO2R online tool was applied to screen out differentially expressed genes (DEGs) between GC and normal gastric samples. Database for Annotation, Visualization and Integrated Discovery was applied to perform Gene Ontology (GO) and KEGG pathway enrichment analysis. The protein-protein interaction (PPI) network of these DEGs was constructed using a STRING online software. The hub genes were identified by the CytoHubba plugin of Cytoscape software. Then, the prognostic value of these identified genes was verified by gastric cancer database derived from Kaplan-Meier plotter platform.

Results

A total of 85 overlapped upregulated genes and 44 downregulated genes were identified. The majority of the DEGs were enriched in extracellular matrix organization, endodermal cell differentiation, and endoderm formation. Moreover, five KEGG pathways were significantly enriched, including ECM-receptor interaction, amoebiasis, AGE-RAGE signaling pathway in diabetic complications, focal adhesion, protein digestion and absorption. By combining the results of PPI network and CytoHubba, a total of nine hub genes including COL1A1, THBS1, MMP2, CXCL8, FN1, TIMP1, SPARC, COL4A1, and ITGA5 were selected. The Kaplan-Meier plotter database confirmed that overexpression levels of these genes were associated with reduced overall survival, except for THBS1 and CXCL8.

Conclusions

Our study suggests that COL1A1, MMP2, FN1, TIMP1, SPARC, COL4A1, and ITGA5 may be potential biomarkers and therapeutic targets for GC. Further study is needed to assess the effect of THBS1 and CXCL8 on GC.

Introduction

Gastric cancer (GC) ranks as the second most common malignancy and is the second leading cause of cancer–related deaths, following lung and bronchus cancer, in China (Chen et al., 2016). Poor 5-year survival in GC is mainly due to the fact that the patients are diagnosed at an advanced stage and even with metastatic diseases and thus miss an opportunity for tumor enucleation (Van Cutsem et al., 2016). Up to date, although biomarkers and therapeutic targets found in GC have made a great improvement to advancing the diagnosis and treatment of GC, the findings associated with GC are not consistent. Therefore, identifying specific biomarkers for the diagnosis and treatment of GC is still urgently necessary (Chakraborty, 2014; Kang et al., 2018; Wadhwa et al., 2013).

In recent years, the Gene Expression Omnibus (GEO) database has provided a powerful tool to elucidate key genetic alternations in carcinogenesis and has extensively used to discover promising biomarkers for cancer diagnosis, new therapeutic target, and prognosis prediction (Cancer Genome Atlas Research N, 2014). Moreover, in order to overcome the inconsistent results attributed to the utilization of either different microarray platforms or a small sample size in research, integrated bioinformatics approaches have been more widely adopted in cancer investigation and a series of valuable bioinformatic reports have been published (Cao et al., 2018; Huang et al., 2018; Li et al., 2018; Liu et al., 2018).

In this work, we firstly performed an integrated analysis and identified differentially expressed genes (DEGs) by using three gene expression profiles between human GC and normal gastric tissue samples from GEO database. Secondly, Gene Ontology (GO) functional enrichment analysis and Kyoto Encyclopedia of Genes and Genomes (KEGG) pathway enrichment analysis were further conducted to analyze the major biological functions of co-modulated DEGs. Then, we constructed protein–protein interaction (PPI) network to identify hub genes related to GC by STRING and Cytoscape. Lastly, the overall survival analyses of the hub genes were validated via “Gastric cancer” database on the online Kaplan–Meier platform.

Materials and Methods

Microarray data source

Using the keyword “gastric cancer” to search on the GEO Datasets database (https://www.ncbi. nlm.nih.gov/geo/), a total of 8,443 series about human gastric cancer (GC) were retrieved from the database. After a careful review, three gene expression profiles on the GEO Datasets database (GSE54129, GSE33651, GSE81948) were downloaded. Among them, GSE54129 was based on Agilent GPL570 platform ([HG-U133_Plus_2] Affymetrix Human Genome U133 Plus 2.0 Array). GSE33651 was based on platform GPL2895 (GE Healthcare/Amersham Biosciences CodeLink Human Whole Genome Bioarray). GSE81948 was based on platform GPL6244 ([HuGene-1_0-st] Affymetrix Human Gene 1.0 ST Array [transcript (gene) version]). All data were freely accessible online, and no experiment on animals or humans was implemented by any of the authors.

Screening for Differentially Expressed Genes (DEGs)

The DEGs analysis between GC and normal tissues was performed using the GEO2R online analysis tool (http://www.ncbi.nlm.nih.gov/geo/geo2r), and the corrected P-value and —logFC— were calculated. Genes that met the cutoff criteria, corrected P-value <0.05 and —logFC— ≥1.0, were considered DEGs. The Venn diagram was carried out for the intersection part via Funrich software (http://funrich.org/).

GO and KEGG pathway analysis of DEGs

The newly powerful enrichment analysis online tool (Enrichr, http://amp.pharm.mssm.edu/Enrichr/) is an essential application for the success of any high-throughput gene function analysis. Gene functions can be classified into biological process (BP), molecular function (MF), and cellular component (CC). KEGG is a widely used database which stores a lot of data about genomes, biological pathways, diseases, chemical substances, and drugs. In current study, we analyzed the DEGs that were significantly up- and downregulated as determined from integrated microarray gastric cancer data, P<0.05 were considered statistically significant.

PPI network construction and hub gene identification

The DEGs were online introduced to the Search Tool for the Retrieval of Interacting Genes (STRING) database (https://string-db.org/) to generate the PPI network. The PPI pairs were extracted with a combined score >0.4 and repeated interactions were removed. Subsequently, the PPI network topology was analyzed by using Cytoscape software (http://www.cytoscape.org/). CytoHubba, a plugin tool in Cytoscape, was applied to calculate the degree of each protein node. Gene with 10 or more gene degree in the PPI network were deemed as hub genes.

Survival analysis of hub genes

The Kaplan–Meier plotter (http://kmplot.com/analysis/) is a web-based tool applied to evaluate the effect of 54,675 genes on survival using 10,461 cancer samples, including 5,143 breast, 1,816 ovarian, 2,437 lung, and 1,065 GC (Lanczky et al., 2016). The Kaplan–Meier plotter GC database was applied to assess the prognostic values of each hub gene in GC patients. The hazard ratio (HR) was given with 95% confidence intervals, and log rank P value was calculated and displayed on the webpage.

Results

Identification of DEGs

Three latest gene expression profiles (GSE54129, GSE33651, GSE81948) were selected in this study. Among them, GSE54129 contained 21 normal samples and 111 GC samples, and GSE33651 included 12 normal samples and 40 GC samples, and GSE81948 included 5 normal samples and 15 GC samples (Table 1). Based on the criteria of P < 0.05 and —logFC— ≥1.0, a total of 3,944 DEGs were screened from GSE54129, including 1,858 upregulated genes and 2,086 downregulated genes. When the GSE33651 dataset was screened by GEO2R, 2,188 DEGs were obtained. Among them, 1,592 genes were upregulated, and 596 genes were downregulated. In gene chip GSE81948, 1,073 DEGs including 525 upregulated genes and 548 downregulated gene were identified. All DEGs were identified by comparing GC samples and normal gastric samples. The DEGs form two sets of sample data included in each of the tree gene expression profiles is shown in Fig. 1. Additionally, Venn analysis was performed to get the intersection of DEGs profiles (Fig. 2). Finally, 129 DEGs were significantly differentially expressed among all three datasets, of which 85 were significantly upregulated genes and 44 were downregulated (Table 2).

Table 1 Details for GEO gastric cancer data.

Reference	Sample	GEO	Platform	Normal	Tumor	
B Liu (2011, unpublished data)	Stomach	GSE54129	GLP570	21	111	
DY Park (2011, unpublished data)	Stomach	GSE33651	GLP2895	12	40	
Canu et al. (2017)	Stomach	GSE81948	GLP6244	5	15	

Figure 1 DEGs between GC samples and normal samples.

(A) The volcano plot for DEGs in GSE54129 data. (B) The volcano plot for DEGs in GSE33651 data. (C) The volcano plot for DEGs in GSE81948 data. x-axes index the log foldchange and y-axes index the -log (P-value). The red dots represent upregulated genes screened on the basis of |fold change| > 1.0, and adjusted P value of <0.05. The green dots represent downregulated genes screened on the basis of |fold change| > 1.0, and adjusted P value of <0.05. The green dots represent genes with no significant difference. FC is the fold change.

Figure 2 Venn diagrams representing the overlaps between three GEO datasets.

(A) Venn diagrams illustrating overlap of upregulated genes in GSE54129, GSE33651, and GSE81948 dataset. (B) Venn diagrams illustrating overlap of downregulated genes in GSE54129, GSE33651, and GSE81948 dataset.

Table 2 Screening DEGs in gastric cancer by integrated microarray.

DEGs	Gene terms	
Upregulated	SLC39A8 S100A10 ANTXR1 DYSF SERPINE2 ADAMTS9 LCP2 NOTCH1 FAS LRP8 C3AR1 MMP3 ANOS1 CALU CXCL10 HAVCR2 MYO1B MMP12 LAMA4 ASAP1 FGD6 HTRA1 ATP1B3 SULF2 FAM83D ANGPT2 CD86 LGALS1 C2 KIRREL PLEK EPHB2 COL15A1 LOX ACTN1 CTSL COL5A1 BCL6 SERPINH1 PMEPA1 NID2 CD248 RAB31 ANGPTL2 ITGB2 MSR1 PCOLCE SPARC IGF2BP3 PLA2G7 PDPN PLXDC2 COL4A2 LAPTM5 FBXO32 CTSK BAG2 MMP2 TNC TNFAIP6 ADAMTS2 OLFML2B COL4A1 CAP2 FNDC1 ITGA5 COL12A1 THY1 SERPINE1 ASPN LY6E THBS1 FN1 GUCY1A3 EDNRA FAP FCGR3B CTHRC1 PDLIM7 SULF1 CXCL8 SFRP4 COL1A1 CHI3L1 COL8A1	
Downregulated	SCGB2A1 SST KRT20 DPCR1 ACER2 SOSTDC1 VSIG1 CAPN9 AKR1B10 ATP4B ADGRG2 TFF2 CWH43 ANXA10 HPGD VSIG2 MAL PSAPL1 PSCA LYPD6B ERO1B ALDH3A1 RASSF6 RDH12 C16orf89 RNASE1 RAB27A AKR7A3 FBXL13 FMO5 ABCC5 PBLD B3GNT6 SELENBP1 TFCP2L1 PXMP2 PDIA2 CAPN13 ALDH6A1 MFSD4A AQP4 SUCLG2 CHGA NRG4	

Functional enrichment analyses of DEGs

GO function and KEGG pathway enrichment analysis for DEGs were performed using the Enrichr. The enriched GO terms were divided into molecular function (MF), biological process (BP), cellular component (CC) ontologies. The results of GO analysis indicated that DEGs were mainly enriched in BP, including extracellular matrix (ECM) organization, endodermal cell differentiation, endoderm formation, wound healing, spreading of epidermal cells and negative regulation of chemotaxis (Fig. 3A). MF analysis showed that the DEGs were significantly enriched collagen binding, peptidase activity, acting on L-amino acid peptides, platelet-derived growth factor binding, low-density lipoprotein particle binding, and serine-type peptidase activity (Fig. 3B). For CC, the DEGs were enriched in endoplasmic reticulum lumen, platelet alpha granule lumen, platelet alpha granule, ruffle membrane, and filopodium membrane (Fig. 3C). These results indicate that most DEGs are significantly enriched in cell proliferation, binding, cell cycle regulation, and transcriptional activity.

Figure 3 GO term and KEGG pathway enrichment analyses performed using Enrichr on DEGs identified from GC samples and normal samples.

(A) The top 10 enriched biological process for DEGs. The horizontal axis represents the number of genes, and the y-axis represents biological process. (B) The top 10 enriched molecular function for DEGs. The horizontal axis represents the number of genes, and the y-axis represents molecular function. (C) The top 10 enriched cellular component for DEGs. The horizontal axis represents the number of genes, and the y-axis represents cellular component. (D) The top 10 enriched KEGG pathway for DEGs. The horizontal axis represents the number of genes, and the y-axis represents KEGG pathway names.

The results of KEGG pathway analysis showed that DEGs were mainly enriched in pathways in ECM-receptor interaction, amoebiasis, AGE-RAGE signaling pathway in diabetic complications, focal adhesion, protein digestion and absorption (Fig. 3D).

PPI network construction and hub gene identification

Proteins interactions among the DEGs were predicated with the STRING online software. A total of 129 nodes and 224 edges were involved in the PPI network, as presented in Fig. 4. The top nine genes evaluated by connectivity degree in the PPI network were identified, as presented in Fig. 5. The results showed that collagen type I alpha 1 (COL1A1) was the most outstanding gene with connectivity degree = 27, followed by thrombospondin 1 (THBS1, degree = 20), matrix metallopeptidase 2 (MMP2, degree = 18), chemokine ligand 8 (CXCL8, degree = 18), fibronectin 1 (FN1, degree = 17), Tissue inhibitor of metalloproteinase 1 (TIMP1, degree = 14), secreted protein acidic cysteine-rich (SPARC, degree = 14), collagen type IV alpha 1 (COL4A1, degree = 13), and integrin alpha 5 (ITGA5, degree = 13). All of these hub genes were upregulated in gastric cancer.

Figure 4 STRING protein-protein interaction network of 85 upregulated and 44 downregulated genes.

The network includes 129 nodes and 224 edges. Circles represent genes, lines represent the interaction of proteins between genes, and the results within the circle represent the structure of proteins. Line color represent evidence of the interaction between the proteins.

Figure 5 Subnetwork of top nine hub genes from protein-protein interaction network using Cytoscape software.

Node color reflects degree of connectivity. The pseudocolor scale from red to yellow represents the top nine hub genes rank from 1–9. Red color represents highest degree, and orange color represents intermedia degree, and yellow color represents lowest degree.

Survival analysis of nine hub genes

To assess the prognostic values of the nine potential hub genes, we subsequently drew overall survival curves (Fig. 6). A total of 876 GC patients were available on Kaplan–Meier plotter platform for the analysis of overall survival. The curves indicated that eight significant DEGs, such as COL1A1, MMP2, FN1, TIMP1, SPARC, COL4A1, and ITGA5, were found to be associated with unfavorable overall survival in GC patients. However, one significant hub gene, THBS1, was found to be not related to over survival in GC patients (Fig. 6B). In addition, one significant up-regulated gene, CXCL8, was found to be associated with favorable over survival in GC patients (Fig. 6D).

Figure 6 Kaplan–Meier overall survival analysis for the top nine hub genes expressed in GC patient samples.

Kaplan–Meier plot of overall survival in subjects with low versus high (A) COL1A1, (B) THBS1, (C) MMP2, (D) CXCL8, (E) FN1, (F) TIMP1, (G) SPARC, (H) COL4A1, and (I) ITAG5 mRNA expression.

Discussion

In the present study, 85 upregulated DEGs and 44 downregulated DEGs between GC and normal human gastric tissues were identified after integrating and screening three gene expression profile datasets from the GEO database. These 129 DEGs were divided into groups by GO functional annotation, including MF, BP, and CC groups. The results of functional enrichment analysis indicated that significant DEGs in GC patients were involved in the GO BP terms such as ECM organization, endodermal cell differentiation, endoderm formation, wound healing, spreading of epidermal cells and negative regulation of chemotaxis. By performing KEGG pathway analysis, 129 DEGs were enriched in the signaling pathways such as ECM-receptor interaction, amoebiasis, AGE-RAGE signaling pathway in diabetic complications, focal adhesion, protein digestion and absorption. A DEG-encoding proteins PPI network was constructed to screen the most closely related genes, and nine hub genes were identified, including COL1A1, THBS1, MMP2, CXCL8, FN1, TIMP1, SPARC, COL4A1, and ITGA5. All of these nine genes were upregulated in GC patients. In addition, the Kaplan–Meier plotter was applied to assess the effect of nine key DEGs on survival in GC patients. Based on the Kaplan–Meier plotter, overexpression of COL1A1, MMP2, FN1, TIMP1, SPARC, COL4A1, and ITGA5 was related to unfavorable prognosis of GC patients. Interestingly, overexpression of THBS1 and CXCL8 was favorable prognostic factors in GC patients, which provide new insights for GC intervention strategy.

Carcinogenesis is a complex process driven by specific genetic alterations and involving multiple signaling cascades. COL1A1, a major component of collagen type I, serves a key role in the tumor cell adhesion and invasion. COL1A1 is overexpressed in a series of cancers including gastric cancer, breast cancer, cervical cancer, non-small lung cancer, and hepatocellular carcinoma. Moreover, COL1A1 overexpression is also related to tumor aggressiveness and poor clinical outcomes (Brooks et al., 2016). The mechanistic study revealed that overexpressed COL1A1 promotes tumor metastasis by regulating the Wnt pathway (Zhang et al., 2018). Furthermore, COL1A1 can be silenced by MiR-129-5p which inhibits gastric cancer cell invasion and proliferation (Wang & Yu, 2018).

THBS1 is an adhesive glycoprotein that mediates cell-to-cell and cell-to-matrix interactions. It has been shown that THBS1 plays essential role in platelet aggregation, angiogenesis, and tumorigenesis (Sajic et al., 2018). Huang and colleagues reported that THBS1 is upregulated by FGF7/FGFR2 growth factors and promotes and knockdown of THBS1 decreased cell invasion and migration (Huang et al., 2017).In this study, the results of Kaplan–Meier plotter indicate that overexpression of THBS1 is not an unfavorable prognostic factor for overall survival in GC patients. Therefore, further study is needed to assess the effect of THBS1 on GC.

MMP2 belongs to the matrix metalloproteinase (MMP) gene family, which are major extracellular enzymes involved in cancer initiation, progression, and metastasis (Yao et al., 2018). The deregulated expression of MMP2 is thought to be involved in multiple pathways disorders causing cancers such as GC cancer, colorectal cancer (Bullock et al., 2013), melanoma (Godefroy et al., 2011), and prostate cancer (Biswas et al., 2010). A meta-analysis showed that MMP2 overexpression might be a potential predictive factor for poor clinical outcome in GC patients (Shen et al., 2014). In addition, increased expression of MMP2 may contribute to the remodeling of the tumor microenvironment in GC (Holmberg et al., 2013).

CXCL8, a member of the CXC chemokine family, is originally described as a mediator of the inflammatory response. Recently, it has been reported that CXCL8 exerts potent pro-tumoral effects in malignant context, such as angiogenesis, cancer stem cells proliferation, and attraction of immunosuppressor cells (Alfaro et al., 2017; Chen et al., 2018). Notably, in this study, it was found that CXCL8 overexpression is related to an improved overall cancer survival in GC patients. Further research is therefore needed to more accurately establish the biological significance of CXCL8 in GC.

FN1 encodes fibronectin, a glycoprotein present in a dimeric form or multimeric form at the surface and in extracellular matrix. The available reports regarding FN1 have mainly focused on cell adhesion and migration processes, blood coagulation, and host defense (Zollinger & Smith, 2017). Recently, many studies have shown FN1 to be associated with a variety of tumor types, and that the expression of FN1 is increased in tumor tissues and cells (Cai et al., 2018; Wang et al., 2017; Yang et al., 2017).

TIMP1 belongs to the TIMP gene family, which are natural inhibitors of the MMPs. TIMP1 is also involved in the cell proliferation, antiapoptotic function, angiogenesis, and oncogenesis (Ries, 2014). Moreover, TIMP1 has been shown to be overexpressed in multiple cancer types and acts as a therapeutic target for GC and other cancers (Omar et al., 2018; Park et al., 2015; Toricelli et al., 2017).

The SPARC gene encodes a cysteine-rich acidic matrix-associated protein, which is required for the collagen and is also involved in extracellular matrix synthesis and promotion of changes to cell shape (Morrissey et al., 2016). In GC, the SPARC expression level is higher compared with normal samples. Moreover, SPARC overexpression is related to reduced overall survival in GC (Nakajima et al., 2018).

ITGA5, a member of the integrin alpha chain family, is known to participate in cell–surface mediated signaling. ITGA5 overexpression has been reported to be in a series of tumors (Chang et al., 2018; Yoo, Kim & Yoon, 2016; Zhao, Wu & Lv, 2015). Increased expression of ITGA5 is also an independent prognostic marker for patients with GC (Ren et al., 2014).

Although we identified several hub genes essential for the progression of GC using comprehensive bioinformatics analysis, several limitations existed in this study. First, this study lacked experimental validation of genes and their functions in GC carcinogenesis. Second, we cannot eliminate the possibility that these key genes may be involved in non-developmental aspects of GC. Finally, the sample size for the RNA-Seq experiments was small, which may have led to potential misleading outcomes. Hence, further bioinformatics analysis of larger samples is warranted, and experimental studies are necessary to corroborate our findings.

Conclusions

In conclusion, we identified some key genes and systematically presented the biological processes and signaling pathways closely related with GC initiation and development. Many of these genes had not previously been reported but could play imperative roles in GC. Notably, compared to other international studies, analyzing prognostic data on CXCL8 and THBS1 separately in our study yielded inconsistent conclusions. Nevertheless, results were inconsistent due partly to unequal sample sizes. Further rigorous studies on CXCL8 and THBS1 are warranted. By sharing genotype data on individuals with GC with fellow investigators, specialists, and other health care professionals, common underlying molecular pathways that may be amenable to potential therapeutic interventions are likely to emerge.

Supplemental Information

Supplemental Information 1 Raw data exported from GSE33651 expression profiling applied for data analyses and preparation for Fig. 1 by using GEO2R online tool

Click here for additional data file.

Supplemental Information 2 Raw data exported from GSE54129 expression profiling applied for data analyses and preparation for Fig. 1 by using GEO2R online tool

NO

Click here for additional data file.

Supplemental Information 3 Raw data exported from GSE81948 expression profiling applied for data analyses and preparation for Fig. 1 by using GEO2R online tool

Click here for additional data file.

Additional Information and Declarations

Competing Interests

Author Contributions

Data Availability

The authors declare there are no competing interests.

Ping Yan and Yingchun He performed the experiments, analyzed the data, prepared figures and/or tables, authored or reviewed drafts of the paper.

Kexin Xie performed the experiments, analyzed the data, prepared figures and/or tables.

Shan Kong performed the experiments, analyzed the data, prepared figures and/or tables, approved the final draft.

Weidong Zhao conceived and designed the experiments, contributed reagents/materials/analysis tools, approved the final draft.

The following information was supplied regarding data availability:

Gene Expression Omnibus: GSE54129, GSE33651, GSE81948.

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
