# Peer review of "In silico analyses for potential key genes associated with gastric cancer"

_PeerJ, doi:10.7717/peerj.6092_

## Round 0.1 · original submission · Minor Revisions

As you can see although all three reviewers provided very favorable comments, some minor concerns were raised. Please address all critical issues indicated by referees and revise you manuscript accordingly.

Reviewer 1 ·

Basic reporting

No Comment

Experimental design

No Comment

Validity of the findings

No Comment

Additional comments

The manuscript entitled ‘In silico analyses for potential key genes associated with gastric cancer’ by Dr. Weidong Zhao and coworkers was on understanding hub genes involved in gastric cancer (GC) metastasis. The aim was to identify the hub genes and investigate the underlying molecular mechanisms of GC. The authors have done an integrated analysis and identified differentially expressed genes (DEGs) by using three gene expression profiles between human GC and normal gastric tissue samples from GEO database. Secondly, Gene Ontology (GO) functional enrichment analysis and Kyoto Encyclopedia of Genes and Genomes (KEGG) pathway enrichment analysis were further conducted to analyze the major biological functions of co-modulated DEGs. Then, constructed protein–protein interaction (PPI) network to identify hub genes related to GC by STRING and Cytoscape. The overall survival analyses of the hub genes were validated via “Gastric cancer” database on the online Kaplan-Meier platform. A total of 85 overlapped upregulated genes and 44 downregulated genes were identified. The study suggests that COL1A1, MMP2, FN1, TIMP1, SPARC, COL4A1, and ITGA5 may be potential biomarkers and therapeutic targets for GC. The authors can also investigate some studies reported by Chakraborty and coworkers Ann Gastroenterol. 2014; 27(3): 231–236. The authors have made a good attempt to understand the hub genes involved in the GC metastasis, present paper is valuable in publishing in PeerJ.

Reviewer 2 ·

Basic reporting

Authors have focused on genes involved in gastric cancer using bioinformatic approaches. Understating the role of key genes that can be potentially used as biomarkers/therapeutic agents is of key significance for cancer diagnostics and potential treatment. Authors have used professional scientific language in the manuscript, however they need to improve phrase construction and need to re check manuscript for spacing and other grammatical errors. Overall the manuscript was very well written only needing some revisions in particular areas. Using GEO, STRING, and cytospace software’s authors have narrowed down the identification of specific genes that might play an important role in gastric cancer. Major weakness in the manuscript is lack of in-depth analysis of the genes identified and the work is similar to several groups with some exceptions in terms of particular hub genes. However the findings are very important and future biochemical research on these particular genes are needed to understand gastric cancer. Conclusion and future directions need to be explained more clearly as identifying genes is one step but what can be done biochemically or clinically is a major step forward.

Experimental design

Authors have chosen bioinformatic approaches for this particular study.
The approaches used for identification of genes using bioinformatic tools are not novel and several other groups have also used similar techniques to identify key genes of potentiality. For examples Li et al used similar approaches based on microarray datasets and found potential hub genes in gastric cancer. Interestingly, the mRNA microarray that they have used are quite different from this particular study. Below is the reference for this particular group.
Li, T., Gao, X., Han, L., Yu, J., & Li, H. (2018). Identification of hub genes with prognostic values in gastric cancer by bioinformatics analysis. World journal of surgical oncology, 16(1), 114.
Authors mentioned in line 109 that they have used 3 microarrays after careful review. On what bases are these particular gene expression profiles were used was not explained very well.
Authors approaches for screening for DEG’s and further use of KEGG and GO analysis is logical in order to get more insight into the function of identified DEG’s. PPI network construction using STRING, hub gene identification, and survival analysis using Kaplan-Meier plotter was very well explained by the authors.

Validity of the findings

Bioinformatic approaches are of significant importance in the field of oncology and identification of potential biomarkers. Findings from the experiments are valid and have great potentiality for future results. However, results and discussion section needs to be elaborated. Figures and tables are represented appropriately showing their results but figure legends can be elaborated. Authors clearly shown the upregulation of key genes when analyzed by PPI network construction. Authors are encouraged to highlight the 7 hub genes from figure 4 with the connection. If possible they can take out some unnecessary nodes and highlight the key interaction. Please consider this as a minor comments. Kaplan-Meier plot results show the prognostic values of identified hub genes.
Authors are encouraged to provide some information on the future research with the identified genes. In future, it is interesting to use protein structure prediction methods and design biochemical experiments on the identified genes. By predicting the tertiary structure, site directed mutagenesis can be performed on the particular genes.
Authors mentioned in Line 265, “Further molecular biological experiments will be performed to confirm the effect of CXCL8 and THBS1 on GC. Chen et al, 2018 group have performed studies on CXCL8 and they have shown that CXCL1 and CXCL8 participate in proliferation, migration process via specific binding with CXCR2. Below is the citation of the article.
Xuyan Chen, Ruifang Jin, Renpin Chen, Zhiming Huang: Complementary action of CXCL1 and CXCL8 in pathogenesis of gastric carcinoma. Int J Clin Exp Pathol 2018;11(2):1036-1045. Int J Clin Exp Pathol 2018;11(2):1036-1045.
THBS1 studies were also done by Huang et al (2017) group showing that THBS1 is upregulated by FGF7/FGFR2 growth factors and promotes and knockdown of THBS1 decreased cell invasion and migration. However these studies are don invitro and further characterization studies are needed.

Additional comments

Authors have intended to identify key hub genes that might be playing an important role in progression of gastric cancer. The manuscript is well written with professional language but need to check for errors and sentence fragmentation.
The methodology and approaches to screen and then narrowing down the results to identify the hub genes is appreciable. However this particular research and study is very similar to some other groups with some different results based on the microarrays that are screened. Interestingly.
Authors need to give an elaborate discussion on how they can use these hub genes as potential biomarkers. They need to address the disadvantages of this approaches as there can be some kind of over interpretation based on the findings. Nevertheless this kind of work is important as it can be used for prediction of gastric cancer progression and the for future research for verification of the DEG’s in gastric cancer.

Reviewer 3 ·

Basic reporting

I recommend the authors to carefully edit the manuscript to avoid typos and avoid errors like spacing between sentences.

All the literature references should be formatted to same style by incorporating volume numbers and initial and final page numbers.

The Figures 3 and 6 should be made more visible by increasing the font size

Experimental design

The manuscript in the present format is addressing a valid point and the research work is in scope of the journal. The authors have described the methods and experiments ver well.

Validity of the findings

The manuscript have novelty and impact in the field

Additional comments

The manuscript entitled "In silico analyses for potential key genes associated with
gastric cancer: describes about understanding hub genes involved in gastric cancer (GC) metastasis that could lead to effective approaches to diagnose and treat cancer.

The manuscript is well written and the conclusions made by the authors are in line with their objectives and experiments.
Hence I recommend accepting the manuscript after a minor revision.

Minor comments:
1. I recommend the authors to carefully edit the manuscript to avoid typos and avoid errors like spacing between sentences.

2. All the literature references should be formatted to same style by incorporating volume numbers and initial and final page numbers.

3. The Figures 3 and 6 should be made more visible by increasing the font size

4. The conclusions of the paper should be rewritten to highlight al points in the paper. The highlights of the paper could be emphasized in conclusions.

---

## Round 0.2 · accepted · Accept

All critical points raised by the reviewers were adequately addressed and the manuscript was revised accordingly. Therefore, the amended version is accepted now.

#